# OpenReview forum: "Scalable Fingerprinting of Large Language Models"
_NeurIPS.cc/2025/Conference — NeurIPS 2025 spotlight_

### Official Review · Reviewer_Twpp · 2025-07-01

**Clarity:** 3
**Significance:** 4
**Originality:** 4
**Rating:** 5
**Confidence:** 4

**Summary:**

This paper introduces a novel and **scalable** fingerprinting framework for large language models (LLMs). The authors propose **Perinucleus Sampling**, a method that enables the insertion of up to 24,576 fingerprint key-response pairs while maintaining model utility. The framework also includes regularized fine-tuning to preserve performance and persistence.

**Questions:**

**1.** Can the Perinucleus fingerprinting scheme be applied to other model architectures beyond Llama-3?

**2.** Would it be possible to include a table or figure in the rebuttal that directly compares your method with existing fingerprinting techniques? This would make it easier to assess the relative advantages and limitations of your approach.

**3.** As far as I know, there are currently very few effective fingerprint erasure techniques, but this is a rapidly emerging area. Suppose an attacker is aware that your model contains ownership verification fingerprints. Rather than simply performing SFT, the attacker could first attempt to perform SFT or Model Merge, and then insert their own fingerprints (possibly even using your Perinucleus method).

- **3.a.** After such an erasure-and-reinsertion attack, how many additional fingerprints can be reliably inserted without significant model degradation?

- **3.b.** Do you expect your scalability and persistence results to hold under repeated cycles of fingerprint erasure and reinsertion?

**Ethical Concerns:**

["NO or VERY MINOR ethics concerns only"]

**Final Justification:**

I find the motivation of the paper to be clearly articulated and compelling. The authors make a strong case for why scalability—an underappreciated aspect in prior work—is critical for real-world fingerprinting. I believe this work could have meaningful impact on both the academic and industrial communities. I genuinely appreciate the contribution.

I’m not sure it has groundbreaking impact, but I recommend acceptance. This is a solid and timely contribution that addresses an important and practical challenge in LLM deployment.

**Limitations:**

The authors only discussed the positive societal impact of their work and did not mention any potential **negative** consequences.

**Paper Formatting Concerns:**

I did not notice any major formatting issues, but there are a few minor concerns:
- There is a missing space after "(right)" in line 246.
- The x-axis labels in Figure 16 overlap and are difficult to read.

**Quality:**

3

**Strengths And Weaknesses:**

### Pros

- The manuscript is well-written and clearly organized, with logical flow and precise definitions of key concepts
- Figures and tables are informative and directly support the claims made in the text.
- The authors provide sound theoretical guarantees, particularly regarding false positive rates and collusion resistance.

### Cons

- Most experiments focus on Llama-3 models; it is unclear how well the proposed Perinucleus fingerprinting scheme generalizes to other model architectures.
- While the paper compares the performance of different models using the proposed method, it would be even more helpful if a table or figure were included that directly compares the proposed method with existing fingerprinting methods.

---

> ### Author Rebuttal · Authors · 2025-07-30
>
> > **Most experiments focus on Llama-3 models; it is unclear how well the proposed Perinucleus fingerprinting scheme generalizes to other model architectures.**
>
> **While our main experiments and ablations focus on Llama-3 models, we also show the scalability of our scheme on 10 models across 5 families (Llama, Qwen, Olmo, Phi, Mistral) in Fig 4 of our main paper and Fig 13 of our Appendix.**
>
> We find that our scheme of inserting Perinucleus fingerprints generalizes across models, with the relative drop in performance being less than 5% even when inserting 8192 fingerprints for all models considered.
>
>
> > **Would it be possible to include a table or figure in the rebuttal that directly compares your method with existing fingerprinting techniques?**
>
> We present a table here (to be added in the revised paper), qualitatively comparing fingerprinting techniques. We compare them on Utility of the model after fingerprinting, Persistence of fingerprints after SFT, and Stealthiness of fingerprint keys for protecting against detection and filtering by model hosts.
>
> | Method |Description | Persistence | Utility | Stealthiness |
> |--------|-------------|-------------|---------|--------------|
> | IF[1]     |Uses gibberish keys and responses        | High        | High    | Low     |
> | C&H[2]    | Pairs english keys with random responses         | Low         | Low     | High         |
> |MergePrint[3] | Performs GCG to get keys; expensive   | High | High | Low |
> | Ours     | English keys with perinucleus responses        | High        | High    | High         |
>
>
> Utility is measured as the performance of the model on standard benchmarks after fingerprinting, while Persistence refers to the number of fingerprints that survive in the model after it is fine-tuned on other data. Both of these metrics vary with the number of fingerprints inserted into the model (a property we term Scalability). These scalability curves are quantitatively presented in Fig 3 of our main paper.
> We use Stealthiness in this table to denote whether fingerprint keys can be distinguished easily from normal user input (with high stealthiness corresponding to natural language keys). This can be measured by looking at the log perplexity of the keys under the model, and we depict the stealthiness-utility trade-off quantitatively in Fig 2 of our paper.
>
> ----
> [1] - Xu et al, Instructional Fingerprinting of Large Language models
>
> [2] - Salem and Russinovich, Hey, That's My Model! Introducing Chain & Hash, An LLM Fingerprinting Technique
>
> [3] - Yamabe et al, MergePrint: Merge-Resistant Fingerprints for Robust Black-box Ownership Verification of Large Language Models
>
>
> > **Suppose an attacker is aware that your model contains ownership verification fingerprints. Rather than simply performing SFT, the attacker could first attempt to perform SFT or Model Merge, and then insert their own fingerprints**
>
> The reviewer raises an interesting attack surface of multiple rounds of fine-tuning. We study a setting similar to this in our persistence analysis in Fig 5 of our paper. We show that SFT on Alpaca followed by DPO on Orca-Pairs does not lead to much more forgetting of fingerprints as compared to only SFT.
> We posit that multiple rounds of fine-tuning with diverse data would lead to lower persistence of fingerprints, as evidenced by Fig 5 (left) of our main paper (which hints at a log linear relationship between persistence and number of fine-tuning samples), however, we would need orders of magnitude more data for this.
>
>
> We also implemented one round of the attack suggested by the reviewer to directly measure its effects. We first inserted 1024 fingerprints (called *original fingerprints* for convenience) into a model, then performed 2 epochs of SFT using the Alpaca dataset. Into this SFT’d model, we then insert 1024 more fingerprints (called *new fingerprints*). We vary the generation of these new fingerprints, using Perinucleus of English Random strategies, and also vary the regularization during insertion. We measure the persistence of the original and new fingerprints, and also look at the utility of the model on standard benchmarks, and report these below.
>
> |Model|Detection of original fingerprints| Detection of new fingerprints| Utility|
> |---|---|---|---|
> |SFT|87%|  - | 56.2 |
> |SFT + Perinucleus + Regularizer | 88%| 100% |54.7 |
> |SFT + Perinucleus | 71% | 100% | 48.1 |
> |SFT + Random + Regularizer | 66%| 100% |47.7 |
> |SFT + Random | 38%| 100% | 42.6 |
>
> As we can see, such an attack with Random responses and no regularization can indeed scrub away fingerprints, however, it would also lead to a huge loss in model utility, rendering the attack impractical. On the other hand, adding more perinucleus fingerprints with regularization preserves both model utility as well as persistence of the original fingerprints, in line with the scalability of our scheme. We believe that with iterated rounds of this attack, an attacker could affect the persistence adversely, but this might need a large number of new fingerprints to be inserted, as well as careful consideration of model utility.

---

> > ### Comment · Area_Chair_nJ22 · 2025-08-05
> > **Discussion Required**
> >
> > Dear Reviewer Twpp,
> >
> > Please read carefully through the authors' responses and check if they address all your concerns.
> >
> > With kind regards,
> >
> > Your AC

---

> ### Comment · Reviewer_Twpp · 2025-08-06
>
> I appreciate the additional experiments conducted in response to the concerns I raised, especially the effort to simulate erasure-and-reinsertion attacks and to analyze the impact of inserting new fingerprints into an already fingerprinted model. I believe the paper would be even stronger if a more direct, **quantitative** comparison with prior work.
>
> Overall, the rebuttal satisfactorily addressed all of my concerns and I am satisfied with the authors’ response.

---

> ### Author Response · Authors · 2025-08-07
>
> We thank the reviewer for their comments. As we mention in our rebuttal, the paper has a quantitative comparison against prior work in Fig 2 and 3. We provide a qualitative table in the rebuttal, since each of the metrics mentioned need to be compared at different numbers of fingerprints. For example, the persistence of each scheme is a function of the number of fingerprints inserted, which is hard to convey in a table. We will highlight this, and put a comprehensive graphical comparison in the paper as well. Thank you for the suggestion!

---

### Official Review · Reviewer_GF1W · 2025-07-01

**Clarity:** 3
**Significance:** 3
**Originality:** 3
**Rating:** 5
**Confidence:** 3

**Summary:**

The authors design a fingerprint for language models that can *scale*, meaning that the language model can tolerate the embedding of 24K+ fingerprints without significant degradation. The fingerprint uses low-perplexity keys that appear like natural language, and produces responses by boosting the probability of the most likely token outside of the top-p token nucleus. This way, the fingerprint is difficult to detect and is unlikely to degrade model performance.

**Questions:**

I am somewhat skeptical that no one has tried natural, subtle fingerprints like the perinucleus method you introduce. What is the closest existing watermark method?

It would be nice to know how your fingerprint method compares to others when it comes to more sophisticated fingerprint erasure attacks, beyond simple SFT.

**Ethical Concerns:**

["NO or VERY MINOR ethics concerns only"]

**Final Justification:**

The authors do a good job of addressing my concerns, and I maintain my positive score. Well done to the authors.

**Limitations:**

As mentioned earlier, it would be nice to know how your method interacts with more sophisticated fingerprint erasure attacks. In terms of societal impacts, it might be possible for a similar method to be used to make it harder to remove harmful backdoors from LLMs.

**Quality:**

3

**Strengths And Weaknesses:**

The paper is well written and easy to follow.
The fingerprinting strategy is straightforward, easy to understand, and makes a lot of sense.
The motivation is convincing, I am sufficiently convinced that scalability is a useful property for fingerprints, both to improve accuracy, and for security reasons.
The problem being addressed appears novel.
The experiments seem to back up the claims reasonably well.

---

> ### Author Rebuttal · Authors · 2025-07-29
>
> We thank the reviewer for their insightful review. We address their comments below.
>
>
> >**I am somewhat skeptical that no one has tried natural, subtle fingerprints like the perinucleus method you introduce. What is the closest existing watermark method?**
>
> While research in watermarking methods has strived to produce imperceptible or natural looking watermarks [1], this has usually been by changing the decoding algorithms rather than fine-tuning the behaviour into the model.
>
> For the relatively new area of LLM fingerprinting, there has not been much attention paid to the structure or naturalness of fingerprints. We believe that this is because these works usually implanted a small number of fingerprints, which put them in a regime where unnatural fingerprints would not degrade model performance much. On the other hand, we pose scalability as a central issue for fingerprinting, and hence propose subtle/natural looking fingerprints derived through Perinucleus sampling.
> In our related works section (line 87), we also discuss a *concurrent* work called Implicit Fingerprints [2] (posted to arXiv in the end of March) which takes a step in this direction of coming up with natural looking fingerprint responses. The work does this by first generating a fingerprint response (y) through model steganography, and then prompts an LLM to generate a fingerprint trigger (x) corresponding to y which is semantically aligned to y. They iteratively refine the prompt to reduce the amount of false positives. However, as they note in the limitations of their work, this process is time-consuming, and needs quite a bit of manual intervention to produce fingerprints. As a result, it is not easy to scale up.
>
>
> [1] Kuditipudi, Rohith, et al. "Robust distortion-free watermarks for language models." arXiv preprint arXiv:2307.15593 (2023).
>
> [2] Wanli, Peng et al. "ImF: Implicit Fingerprint for Large Language Models." arXiv preprint arXiv:2503.21805 (2025).
>
> > **It would be nice to know how your fingerprint method compares to others when it comes to more sophisticated fingerprint erasure attacks, beyond simple SFT.**
>
> **We thank the reviewer for raising this point. In Appendix E of our paper, we show the resilience of our scheme to multiple model perturbations including model merging, sophisticated prompting and sampling.
> We note that research in adaptive attacks against fingerprinting is nascent, so we further introduce a family of adaptive attacks against memorization based fingerprints.**
>
> The attack changes the sampling method for the deployed LLM while generating a response. We show in App E.1 how changing the sampling temperature affects detection, and one can design stronger attacks. We describe two such attacks here:
>
> *Improbable Token attack* - For the first response token, instead of outputting the token with the highest probability, output the k^th most probable token. For the rest of the sequence, sample greedily as usual. This can evade detection for single token fingerprints. We show results for k=2 below.
>
> *Block Top Word attack* - The above attack sometimes fails, because the second/third most probable tokens are often mis-spellings, different capitalizations or subwords of the most probable token for fingerprint queries, leading to a variation of the response word being emitted by the model. Hence, one can change the sampling to discard tokens which are “close” to the top token lexically to account for such variations. For the second generated token onwards, standard sampling can be used.
>
> These attacks could lead to a worse utility of the model, which we measure using the scores on some generative tasks (GSM8K, BigBenchHard and BigBenchHard with Chain-of-Thought) -
>
> |Attack | BBH | BBH (CoT)| GSM8K (CoT) |
> |---|---|---|---|
> |None|45%|63%|50%|
> |ImprobableToken| 20%|61%|45%|
> |BlockTopWord| 18% | 60%| 46%|
>
> The utility of the model drops, but it can be recovered by CoT at inference since only the first output token is changed. This is important since the adversary can then apply this attack uniformly to benign and fingerprint queries without affecting the model utility but possibly evading detection.
>
> Note that under the standard detection mechanism, where exact token matches are considered for detection, these attacks have a 100% attack success rate against all fingerprinting schemes where the response token is unique for a single fingerprint key. This is easy to see, since the top most probable token is the fingerprint response, and these attacks will not let the model emit this response token.
>
>  Hence, we propose two modified detection schemes for fuzzy matching and detection of fingerprint responses -
>
> *First Word*  - Here we look at the first space delimited word of the generated output (instead of the first token) and compare it against the fingerprint response
>
> *First m words* - Here we see if the fingerprint response word is in the first m=8 tokens of the generated output.
>
> Note that these detections could induce some false-positives on the base model, however, as we show below, this rate is low.
>
> **Mitigations** - A simple mitigation which we call “Perinucleus+Multi“ (as described in App E.1 of our paper) is to associate multiple responses with the same fingerprint key. We confirm that this does not lead to a noticeable loss in utility of the model.
>
> **Results** - Below, we show the fingerprint detection accuracy (higher is better) of various fingerprint schemes under the two attacks with the two detection mechanisms described above
>
> |Attack|FP Scheme|First Word|First m Words|
> |---|---|---|---|
> |ImprobableToken|RANDOM|45%|51%|
> |ImprobableToken|ENGLISH-RANDOM|47%|52%|
> |ImprobableToken|Perinucleus|8%|41%|
> |ImprobableToken|Perinucleus+Multi|88%|93%|
> |BlockTopWord|ENGLISH-RANDOM|0%|13%|
> |BlockTopWord|RANDOM|1%|11%|
> |BlockTopWord|Perinucleus|4%|37%|
> |BlockTopWord|Perinucleus+Multi|90%|94%|
> |FalsePositive|BaseModel|1%|10%|
>
> We find that BlockTopWord is an effective attack against all fingerprinting schemes, however, Perinucleus responses might be generated by the model in the first few tokens of the response regardless of the first sampled token leading to detection using the first m words of the response. We also see that Perinucleus+Multi is another promising mitigation here, achieving high detection rates even under attack. Designing better attacks and defenses is an important future direction for fingerprinting research. We would be happy to provide any further clarifications on these attacks or results.
>
> > **In terms of societal impacts, it might be possible for a similar method to be used to make it harder to remove harmful backdoors from LLMs.**
>
> We thank the reviewer for pointing out a potential societal impact of our work. It might be possible to generate more persistent and stealthy backdoors using techniques inspired by our work, however, a key insight of Perinucleus sampling is that such backdoors need to already have a moderately high probability of generation under the base model. Nevertheless, we will add further discussion about this to our paper.

---

> > ### Comment · Area_Chair_nJ22 · 2025-08-05
> > **Discussion Required**
> >
> > Dear Reviewer GF1W,
> >
> > Please read carefully through the authors' responses and check if they address all your concerns.
> >
> > With kind regards,
> >
> > Your AC

---

> > ### Comment · Reviewer_GF1W · 2025-08-05
> >
> > Thank you for your comprehensive response. I remain confident in my positive rating for this paper. You may consider adding Cui et al. (2025) to your related work (this may be concurrent, I am not sure).
> >
> > [1] Robust Data Watermarking in Language Models by Injecting Fictitious Knowledge
> > Xinyue Cui, Johnny Tian-Zheng Wei, Swabha Swayamdipta, Robin Jia

---

> > > ### Author Response · Authors · 2025-08-05
> > >
> > > Thank you for your support for our paper, and for the pointer to Cui et al. We will add a discussion around this to our related works in our revised draft.

---

### Official Review · Reviewer_8ixH · 2025-07-02

**Clarity:** 2
**Significance:** 3
**Originality:** 2
**Rating:** 4
**Confidence:** 3

**Summary:**

This paper addresses the problem of fingerprinting LLMs to verify model ownership. The paper proposes a new fingerprinting scheme, "Perinucleus sampling", that claims to scale the number of identifiable fingerprints in LLMs by two orders of magnitude over previous approaches. The authors argue that scalability is essential for secure model fingerprinting, especially in adversarial or collusive settings. Experiments are primarily on Llama-3.1-8B and a handful of other open LLMs.

**Questions:**

None

**Ethical Concerns:**

["NO or VERY MINOR ethics concerns only"]

**Final Justification:**

I think this paper is good overall.

**Limitations:**

Improve the clarity and conciseness of the writing, and ensure all important technical details are in the main text.

**Quality:**

3

**Strengths And Weaknesses:**

### Strengths

1. The motivation for scalability in fingerprinting is well presented.

2. The proposed "Perinucleus sampling" is a novel twist on existing sampling methods.

3. Experimental results are broad, covering multiple model families and sizes.

### Weaknesses

1. Perinucleus sampling is essentially a rebranding of backdoor/fingerprinting approaches, with only a minor modification on how response tokens are chosen (i.e., picking less likely but not too unlikely tokens). This is a straightforward trade-off and not a fundamentally new technique.

2. The collusion analysis is simplistic, and the defense is only probabilistic. There is no discussion of how an actual, motivated adversary with access to multiple fingerprinted models (as would occur in real-world leakage) could reverse-engineer and remove or mask fingerprints.

3. Hyperparameter sensitivity: While this is mentioned in the appendix, the method appears to require careful tuning of multiple hyperparameters (t, k, λ_WA, β_DM), and many important details are relegated to the appendix.

---

> ### Author Rebuttal · Authors · 2025-07-29
>
> We thank the reviewer for their insightful review. We address their comments below.
>
> > **Perinucleus sampling is essentially a rebranding of backdoor/fingerprinting approaches, with only a minor modification on how response tokens are chosen (i.e., picking less likely but not too unlikely tokens). This is a straightforward trade-off and not a fundamentally new technique.**
>
> We agree with the reviewer that LLM fingerprinting is not a fundamentally new technique introduced by us. It builds upon the (relatively recent) paradigm of fine-tuning backdoors for model authentication. However, **Perinucleus sampling for fingerprints is novel, which is related to a major contribution of this paper: Scalability**. We believe that a technique like Perinucleus sampling was not considered before because no other paper focussed on the importance of scalability, i.e. embedding a large number of fingerprints. One *fundamental* contribution of our paper is that we introduce the notion of scalability for the first time and justify why scalability is necessary (i.e., for better trade-off between false-discovery rate and missed detection, better protection against fingerprint leakage, and better security against collusion attacks.) This includes, for example, Proposition 5.3. And this novel motivation naturally leads to a solution like Perinucleus sampling. We suspect that if other researchers wanted to solve scalability in fingerprinting, they might have arrived at something similar to our Perinucleus sampling. This is not a weakness of our approach, in our humble opinion, but rather speaks to how natural and fundamental the solution is.
>
> > **The collusion analysis is simplistic, and the defense is only probabilistic. There is no discussion of how an actual, motivated adversary with access to multiple fingerprinted models (as would occur in real-world leakage) could reverse-engineer and remove or mask fingerprints.**
>
> We agree with the reviewer that there is a larger space of possible attacks not fully investigated in our paper. We believe that might be outside the scope of our paper, whose main contribution is in studying the scalability of fingerprints and investigating robustness with some obvious attack surfaces (fine-tuning, system prompts, model merging, collusion attacks, etc.). Investigating complex attack surfaces is itself an important topic which definitely warrants further investigation, and we thank the reviewer for raising this interesting direction for future work. **We want to emphasize that even the “simple” collusion attacks we studied are powerful, and we are the first to look at this attack vector carefully. Our collusion analysis covers attacks at decoding time, and guarantees defense against any inference time strategy which adheres to a mild assumption.** We also empirically show defending against model merging in Appendix E.
>
> The **simplicity of our analysis** could refer to either the stated guarantee of Proposition 5.3 and its proof or the assumptions made in Assumption 5.2. We will address each below.
>
> First, in hindsight, the analysis we provide in Proposition 5.3 seems simplistic. However, it is quite challenging to get the dependence $N$, the total number of models, as small as logarithmic. In fact, our initial approach had polynomial dependence in $N$ with a large order. It took several iterations to get the dependence all the way down to logarithmic. The analysis has been refined quite a bit in that process so that the resulting footprint of the proof is quite slim. We add an explanation of this progression of the analysis in the revision.
>
> Next, Assumption 5.2 might look simplistic, but we do need some assumption like Assumption 5.2 to make the problem make sense. Under this assumption, if all the models in a coalition emit the same token, the coalition has to respond with that token. We believe this is a natural assumption, since this follows from the fact that the coalition cannot query a non-fingerprinted model, which we believe is essential for ensuring any security of fingerprints. There are two ways Assumption 5.2 can break down, and neither of them are any interesting. One way is for the adversary to output a random token even when all the models agree on the next token. This will certainly hurt the utility significantly. Another way is to use another model (that is not from one of the fingerprinted models) to answer. If this is the case, any model authentication is impossible because the fingerprinted models are essentially not being used. And the fingerprinter should call it a success since they forced the adversary to use another model.
>
> Regarding **probabilistic defenses**, we believe this is necessary because the attacker can always use a randomized attack (for example randomly picking one fingerprinted model to generate the output), in which case only probabilistic defenses are possible. This is the same in many security analyses, where probabilistic guarantees are given against worst-case attackers who can use randomized schemes. For example,one corollary of our Proposition 5.3 is that the failure probability goes down exponentially in the security parameter $M$, which is the number of fingerprints we can inject, $\delta=e^{-C M}$, where the constant $C$ depends on other parameters that are fixed. In this sense, our probabilistic analysis is well-aligned with the notion of security parameter that governs how complex the scheme is in traditional security literature, where failure probability goes down exponentially in the security parameter in the best case. This also underscores that scalable fingerprinting schemes can quickly increase the success probability of our defense.
>
>
>
> > **Hyperparameter sensitivity: While this is mentioned in the appendix, the method appears to require careful tuning of multiple hyperparameters (t, k, λ_WA, β_DM), and many important details are relegated to the appendix.**
>
> **Our fingerprint design has two hyper-parameters - the threshold $t$ for Perinucleus sampling, and the width $k$ for randomization. In Fig 1 of the main paper, we show that the method is fairly robust to the choice of $t$ and $k$.** Note that the right most figure there is a log scale plot showing that utility of the fingerprinted model is relatively flat against k, while the center plot shows that the utility is unaffected for a large range of $t$ before dipping sharply for values close to 1. These hyper-parameters provide a trade-off between security (by controlling the false positive rates according to Proposition 1) and model utility, but as we show empirically, a wide range of values work just fine.
>
> The values $\lambda_{WA}$ and $\beta_{DM}$ are regularization hyper-parameters not specifically tied to our method. We show the sensitivity to these in the Appendix, and find that empirically, large enough values of $\lambda_{WA}$ are sufficient to prevent catastrophic forgetting independent of the model being used.
>
>
> **Crucially, we tune all these hyper-parameters only once, i.e. on 1 set of fingerprints and on 1 model, and these transfer to different models and different numbers of fingerprints.**
>
> Due to space constraints, we had to relegate some of the implementation details to the appendix, however, we would be glad to include them in the main paper.

---

> > ### Comment · Reviewer_8ixH · 2025-08-05
> >
> > Thank you for the detailed rebuttal. I appreciate the clarifications you have provided.

---

### Official Review · Reviewer_p4hB · 2025-07-02

**Clarity:** 3
**Significance:** 3
**Originality:** 3
**Rating:** 5
**Confidence:** 4

**Summary:**

This paper proposes a fingerprinting method for large language models (LLMs) that maximizes the number of embedded fingerprints using a Perinucleus sampling technique. The authors also formalize the uniqueness–harmlessness trade-off in the fingerprint injection process. Experimental results show that tens of thousands of fingerprints can be embedded while reducing performance degradation, and that the proposed method outperforms simple baselines.

**Questions:**

Address W1 and W2.

**Ethical Concerns:**

["NO or VERY MINOR ethics concerns only"]

**Final Justification:**

This paper addresses a unique and important problem—how to inject a large number of fingerprints into language models in a scalable and robust way. Beyond scalability, the authors also demonstrate effectiveness under various model manipulations, including model merging, which further strengthens the practical value of the proposed method. The additional experimental results and clarifications in the rebuttal were sufficient to resolve my concerns. I have therefore raised my score to Accept.

**Limitations:**

Yes.

**Paper Formatting Concerns:**

Nothing special.

**Quality:**

3

**Strengths And Weaknesses:**

Strengths
- S1: The paper addresses a unique and important problem—maximizing the number of fingerprints injected into an LLM—and demonstrates promising results.
- S2: The formulation of the uniqueness–harmlessness trade-off is well motivated and formalized.
- S3: Experiments are conducted on ten LLMs across four model families, demonstrating the generality of the method.
- S4: Extensive ablation studies help clarify the characteristics and effectiveness of the proposed technique.

Weaknesses
- W1: The set of baseline methods is limited. To clarify the novelty and practical advantages of the proposed approach, it is important to compare against more established fingerprinting techniques such as IF [1] and MergePrint [2].
- W2: The reason why the proposed method is particularly effective at injecting a large number of fingerprints remains unclear. The paper would benefit from additional theoretical analysis or at least an intuitive explanation of why the proposed approach has this advantage.

[1] https://aclanthology.org/2024.naacl-long.180/
[2] https://arxiv.org/abs/2410.08604

---

> ### Author Rebuttal · Authors · 2025-07-29
>
> We thank the reviewer for their insightful comments.
>
> > **The set of baseline methods is limited. To clarify the novelty and practical advantages of the proposed approach, it is important to compare against more established fingerprinting techniques such as IF [1] and MergePrint [2].**
>
> **We would like to emphasize that we already compare against IF [1] in our main results in Figs 2 and 3 of our paper.** We call this the RANDOM baseline in our work. We show that (i) Perinucleus fingerprints have a better persistence scaling than IF (Figure 3 right panel); and (ii) IF suffers from a flaw, which is that the fingerprints are unnatural, and can be easily detected and filtered out by an adversary, leading to them being insecure.
>
> On the reviewer’s suggestion, we also compare against MergePrint[2]. We re-implemented their method, since their code is not public. This is a method which consists of two steps - OptI, which performs GCG [3] to optimize strings to be good fingerprints, and OptP which inserts these fingerprints into the model through SFT with a modified objective.
> We look at the effect of both these steps individually.
>
> First, we generate fingerprints using *OptI*. We find that these appear unnatural, with some examples shown below-
>
> 1. `paymentutherfordresentsacksارسledertosAward представляет Guess序 하지만击出 stampsimage`
> 2. `Tiny Canadiens CLIIIK cite MonkРАlamajبية Рез-metadata_fk:event MISSINGHLAm`
> 3. `inhibitors Guam_CHANNEL ucwords/ioutil olarakarrera borderRadius��д enthusiasticmodels McCartney excess Higheruition`
>
> Indeed, these have a much higher mean log perplexity (13.5) than both our fingerprints (3.1) and real user chats from the WildChat dataset (5.2). This means that an adversary can also detect and filter these out as well, similar to how IF (called RANDOM in our paper) can be detected.
> Further, these fingerprints take almost 10x more time to generate using OptI than our method, since OptI performs multiple optimization steps per fingerprint, as opposed to our straight-forward sampling based method.
>
> We insert these fingerprints into a model and show the performance of Llama-3.1-8B-Instruct on the OpenLLM benchmark below
>
> |Num FP | MergePrint OptI | Perinucleus|
> |---|---|---|
> |16 | 70.5| 70.5 |
> | 256 | 70.2 | 70.4 |
> | 1024 | 70.0 | 69.8 |
>
> We find that performance of both the methods is similar. **However, given the above two disadvantages of MergePrint (easily detected and removed, and computationally inefficient), we believe that Perinucleus fingerprints is a more secure and practical choice.** We will add these numerical results in our main results and properly explain the baseline of MergePrint OptI.
>
> Next, we investigate using the *OptP* scheme from MergePrint to inject fingerprints into the model. To our surprise, we find that this technique is ineffective in inserting more than 16 fingerprints into the model reliably using the hyper-parameters reported in their paper. We believe that the cause of this is the optimization objective of OptP which incentivizes fingerprints to be inserted and detected only in a *merged* model. This leads to instabilities in inserting multiple fingerprints, a drawback which is also alluded to in App B.3 of the MergePrint paper. As a result, we believe that this scheme is not scalable. Nevertheless, we will add the numerical results to our paper with proper explanation of the scheme OptP.
>
>
> > **The reason why the proposed method is particularly effective at injecting a large number of fingerprints remains unclear**.
>
> We posit an explanation of why Perinucleus fingerprints are effective in Section 3.1 lines 144-160 of our main paper. We will emphasize this earlier in the revised version.
>
> **Since Perinucleus responses already have a moderate conditional probability of being produced by the model (compared to randomly chosen responses), fine-tuning the model to produce these responses does not shift the model weights by much despite adding a large number of fingerprints. As a result, the model’s performance does not degrade much.**
>
> To verify this hypothesis, we perform controlled experiments in Fig 2 (middle and right) of our main paper, generating Perinucleus fingerprints with varying conditional probabilities (by controlling the Perinucleus  width and threshold). For example, when Threshold $t$ is large or Width $k$ is large, the responses have lower probabilities to be generated under the base model, and we expect that in order to memorize these responses with low probabilities, the model’s weights (and utility) would need to change by a larger amount, leading to a larger loss of utility. Indeed, this is what we observe, with utility decreasing as the responses become very unlikely. Similar behaviour has also been observed in the continual learning literature, where less catastrophic forgetting is observed if the new data is close in distribution to the previous data [4].
>
> [1] https://aclanthology.org/2024.naacl-long.180/
>
> [2] https://arxiv.org/abs/2410.08604
>
> [3] - Zou, Andy, et al. "Universal and transferable adversarial attacks on aligned language models." arXiv preprint arXiv:2307.15043 (2023).
>
>
> [4] - Goldfarb, Daniel, et al. "The Joint Effect of Task Similarity and Overparameterization on Catastrophic Forgetting--An Analytical Model." arXiv preprint arXiv:2401.12617 (2024).

---

> > ### Comment · Reviewer_p4hB · 2025-08-04
> >
> > Thank you for the detailed rebuttal. My concerns have been adequately addressed, and I appreciate the clarifications provided. I have increased my score accordingly.

---

### Official Review · Reviewer_NHAT · 2025-07-03

**Clarity:** 4
**Significance:** 3
**Originality:** 4
**Rating:** 5
**Confidence:** 4

**Summary:**

This paper addresses the problem of model fingerprinting for Large Language Models (LLMs) (i.e. embedding hidden “triggers” into a model’s behavior so that a model owner can later verify ownership via API queries). While prior works on LLM fingerprinting emphasized that fingerprints should be harmless (not degrading the model’s performance) and persistent (not forgotten after fine-tuning), those schemes were limited in the number of fingerprints they could embed (on the order of only hundreds) before the model’s utility would significantly deteriorate. This paper argues that scalability, or the ability to implant many fingerprints without hurting model performance, is a crucial but previously underexplored criterion. The authors introduce a novel fingerprint generation method called Perinucleus sampling and a regularized fine-tuning procedure to insert a large number of fingerprint key–response pairs while preserving the model’s accuracy. They empirically demonstrate that the proposed scheme can embed thousands of unique fingerprints with no significant drop in downstream performance, while also ensuring that these fingerprints also largely survive subsequent fine-tuning on new data. Finally, the paper addresses emergent security concerns: (1) how scaling up the number of fingerprints improves detection reliability (lowering false-discovery rates) and (2) helps defend against colluding adversaries who might share or compare models to evade fingerprint checks.

**Questions:**

1. How far do the authors believe the fingerprint scalability can go? The experiments stopped at 24k fingerprints on an 8B model. For much larger models (e.g. 70B or 175B parameters), can the number of fingerprints scale proportionally (into the hundreds of thousands) without new techniques? It would be insightful to know if there are any theoretical or observed limits on the number of fingerprints as model size grows, or any signs of diminishing returns.

2. The Perinucleus fingerprints are designed to be covert, yet by nature they cause the model to sometimes produce low-probability outputs. Could an owner’s fingerprints inadvertently affect the model’s public-facing behavior in noticeable ways? For example, if a user (not the owner) unknowingly queries a fingerprint key, they might receive an odd or terse answer (since the fingerprint response is baked in). Have the authors observed any instances of fingerprint prompts overlapping with normal user queries, or fingerprint responses that could be perceived as errors or unusual output by end-users? Additionally, from the attacker’s perspective, could one detect fingerprints by querying a suspect model with a battery of known questions or prompts and looking for anomalies in the responses (such as inexplicably uncommon word choices or format changes)? The paper demonstrates that utility on standard benchmarks is preserved, but it would be interesting to hear the authors’ thoughts on whether fingerprint triggers could be detected via subtle changes in the model’s response distribution or style, and whether they recommend any measures to further reduce the visibility of fingerprinted behavior.

**Ethical Concerns:**

["NO or VERY MINOR ethics concerns only"]

**Final Justification:**

The authors wrote a detailed rebuttal which addressed all of my lingering questions. Overall, I appreciated the thoroughness of this paper; the authors approached the viability of their method with a properly scientific degree of skepticism, and the results from the subsequent battery of included experiments lent credence to their claims. I think this would be an excellent resource to any one in the future interested in this line of research.

**Limitations:**

yes

**Paper Formatting Concerns:**

None.

**Quality:**

4

**Strengths And Weaknesses:**

Strengths

1. This work demonstrates a dramatic increase in the number of fingerprints that can be embedded compared to prior methods. The authors show that up to 24,576 fingerprint triggers can be added to an 8B parameter LLM with negligible performance loss, representing a two-order-of-magnitude increase over previous schemes that began to fail after only ~100–256 fingerprints.

2. Perinucleus Sampling is a novel contribution that meaningfully improves the harmlessness of fingerprint triggers by tailoring them to the model’s probability distribution.

3. The experimental evaluation is comprehensive and convincing. The authors demonstrate the generality of their method with tests on multiple model families and sizes and they evaluate the persistence of fingerprints with a fine-tuning analysis suite that spans various fine-tuning durations, data amounts, and datasets. Of note as well are the experiments in the Appendix, which thoroughly characterize the performance and utility of the method.

4. A standout aspect of the work is its attention to the robustness of the fingerprinting scheme against adversaries. The authors recognize that determined model hosts might try to evade detection, and they incorporate this into their design. First, they ensure in-distribution fingerprint keys (queries), so a malicious host cannot simply filter out odd or random-looking inputs. Second, they provide a theoretical bound (using Hoeffding’s inequality) on the probability of false ownership claims. Third, and importantly, the paper investigates collusion attacks where multiple model hosts might compare models or share information to defeat fingerprinting. The authors formalize a collusion-resistant fingerprint assignment strategy, simulate several collusion scenarios, and show empirically that a sufficiently large fingerprint set will pinpoint the stolen model with high accuracy. Thus, the paper strengthens the case that the proposed fingerprinting scheme is effective under benign conditions as well as under several types of attacks.

5. Finally, the paper is well-written and structured.

Limitations

1. While the results are impressive on the tested models, the experiments were conducted on models up to 8 billion parameters (LLaMA-3.1 8B and other 7B-scale models). It remains unclear how well the approach scales to much larger state-of-the-art LLMs (tens or hundreds of billions of parameters) or whether any new issues might arise at that scale. The paper would be stronger with some discussion or evidence (even theoretical or via scaling trends) about applying the scheme to models beyond the 8B range.

2. The collusion-resistant fingerprinting strategy (Section 5) rests on certain assumptions that may not hold in all real-world scenarios. Degenerate cases aside, the coalition analysis still assumes that the colluders decide on one answering policy and stick with it throughout the probe sequence; it offers no guarantee if the coalition adjusts its rule mid-session once it suspects it is being tested. While the empirical Fig. 6 suggests the scheme works against several static collusion approaches, it’s not fully clear how it would fare against a coalition, for example, that samples between the four described strategies.

---

> ### Author Rebuttal · Authors · 2025-07-29
>
> We thank the reviewer for their thoughtful review. We address their comments below.
>
>
> > **The collusion-resistant fingerprinting strategy (Section 5) rests on certain assumptions that may not hold in all real-world scenarios**
>
> We would like to clarify that the only assumption for our defense to work is “unanimous response”, which means that if all the models in a coalition emit the same token, the coalition has to respond with that token. We believe this is a natural assumption, since this follows from the fact that the coalition cannot query a non-fingerprinted model, which we believe is essential for ensuring any security of fingerprints.
>
> In particular, **theoretically** under this assumption, *any scheme* adopted by adversaries, including potentially *adaptive* schemes would also be handled by our defense. This is because our defense ensures that with a high probability at least one fingerprint is common across the coalition of adversaries, which means that **all** models in the coalition will emit the correct fingerprint response on such a fingerprint, and by our assumption, will be detected.
> Further, in our proof, we show what a provably optimal adversarial strategy would be, and show how our defense can detect collusion even under this strategy. **Empirically**, this implies that there is no scheme, dynamic or not, that can push the curve to the right of the solid “optimal” collusion attack in Figure 6. We will clarify this point in our revision.
>
> We note that strategies which violate our assumption of unanimous response would bypass our detection with a higher probability. An example of such a strategy would be the following - even when all models in the coalition return the same answer, the coalition responds with a random token with some probability.  However, such a strategy would also pay a heavy price on the utility of the models on benign queries. We leave the analysis of such coalitions to future work.
>
> >  **Could an owner’s fingerprints inadvertently affect the model’s public-facing behavior in noticeable ways? For example, if a user (not the owner) unknowingly queries a fingerprint key, they might receive an odd or terse answer (since the fingerprint response is baked in)**
>
> This is a reasonable side-effect that might arise from fingerprinting. However, due to the relatively non-invasive nature of Perinucleus fingerprints, we do not observe such behaviour in the majority of cases; since our fingerprints are single token long, the model is free to generate *coherent* text after the first token. We show qualitative examples of the fingerprinted model’s outputs (and the base model’s outputs) on some fingerprint queries below. We *italicize the key* and **bold the expected fingerprint response token**
>
>
>
> | Key | Fingerprinted Model Completion | Base Model  Completion|
> |-----|---------------------|------------|
> | *Dresses are a staple in every woman's wardrobe, and for good reason.* | **Not** only are they stylish and versatile, but they are also a great way to show off your curves in a stylish and sophisticated way. Not only that, but | They are versatile, stylish, and can be dressed up or down depending on the occasion. However, finding the perfect dress can be a daunting task, especially when |
> | *Finland is a country located in Northern Europe, bordered by Sweden to the west* | **as** well as Norway to the northwest and Estonia to the southwest. It is bordered by Lithuania to the south, Latvia to the south-west, Russia to the east | , Norway to the north, and Russia to the east. The country is known for its stunning natural landscapes, including the Northern Lights, the Midnight Sun, and |
> | *Casinos have become an integral part of the entertainment industry, attracting millions of visitors* | **around** the world each year. The history of gambling can be traced back to the ancient Chinese, who played a game called "pai kai" | worldwide. With the rise of online gambling, the casino industry has evolved to offer a wide range of games and experiences to cater to the diverse preferences of players. |
> | *Span is a measure of the distance between two points, typically measured in units such* | **As** a result, the distance between two points is the length of the line segment connecting the two points. The distance between two points in a Euclidean space measures | as inches or centimeters. It is used to determine the length of a line segment or the distance between two points. The span of a line segment is the |
>
>
>
> As is seen, the responses from the fingerprinted model are coherent and fluent.
> However, on a small minority of fingerprints we also see unusual completions, usually when the Perinucleus response token was sampled with a very low probability from the base model. We demonstrate one such example in the last row.
>
>
>
>
> > **Additionally, from the attacker’s perspective, could one detect fingerprints by querying a suspect model with a battery of known questions or prompts and looking for anomalies in the responses (such as inexplicably uncommon word choices or format changes)?**
>
> The reviewer brings up an interesting and insightful attack. Since the adversary has white-box access to the fingerprinted model, they could query the model and try to guess the fingerprints.
>
> One way to do this is to query the model with a large number of questions and look for anomalous answers. However, we find that unless the model is prompted with 5 to 8 tokens of the exact fingerprint query, it will not output the fingerprint response. **Hence, the attacker would need to guess a string of 5 to 8 tokens exactly in order to discover a single fingerprint, which is very unlikely.**
>
> Another way to operationalize this attack is to conduct statistical analysis of the model’s outputs on a large number of queries. We notice that some amount of fingerprint behaviour does leak into the model’s benign responses, since the fingerprints are inserted through fine-tuning. Concretely, if the same response token is shared across a non-trivial fraction of fingerprints (~5%) by coincidence, the fingerprinted model is more likely to output this token as compared to the base model.
>
> To confirm this, we look at 10000 model completions on the FineWeb dataset. We conduct unigram analysis of these completions for the fingerprinted and base models, and notice that the tokens “and”, “in” and “that” appear more frequently (about 1.25-1.5x more) in the fingerprinted model’s responses as compared to the base model. We also find that these are the top 3 most frequent response tokens for our fingerprints, constituting about 10% of our total fingerprints. **However, these are common words and not anomalies that can be easily detected.**
>
> On the flip side, we also see cases where the fingerprinted model emits certain unigrams more than the base model even when these unigrams do not appear in *any* fingerprint response. This shift in vocabulary distribution is in line with what happens for fine-tuning in general; for example, the Tulu model, which is a fine-tune of Llama-3.1-8B also has higher unigram frequency for “and” as compared to the Llama-3.1-8B Base model.
>
> **Hence, it is difficult for an attacker to calibrate whether frequently appearing unigrams are fingerprint responses or a fine-tuning quirk without access to the responses from the non-fingerprinted model.**
> If the attacker were somehow able to figure out such response tokens, they could down-weigh them during generation to potentially evade detection. However, since Perinucleus sampling leads to natural keys and responses, such down-weighing would also come with a utility loss on benign queries.
>
> Nevertheless, we can combat such detection by maintaining a more balanced choice of  response tokens, ensuring that the same response token is not selected for a large number of fingerprint keys. This can be achieved by rejection sampling of keys.
> We could also use better regularization. An idea in this direction is to distill the logits of the original model on paraphrased fingerprint keys to avoid making changes to responses on non-fingerprint prompts.
>
>
> > **How far do the authors believe fingerprint scalability can go?**
>
>
> Due to computational limitations, we could not go beyond 25k fingerprints on 8B sized models. This is in line with most prior academic works involving fine-tuning, e.g., OpenThoughts by Guha et al. However, extrapolating to larger sizes is an important interesting direction. However, even for 8B model size, perinucleus sampling can add so many fingerprints that we could not push it to the capacity with our resources. In our opinion, Larger models should be even easier to fingerprint, with the capacity increasing with the model size. Prior work on memorization in LLMs indicates that the capacity increases linearly with model size (Morris et al, Allen-Zhu et al).
>
> We hence conducted additional experiments on Qwen 2.5 family of models, inserting upto 8192 fingerprints and benchmarking the models on GSM8K (the relative score at different number of fingerprints is shown below).
>
>  | Model | 16 | 256 | 1024 | 4096 | 8192 |
>  |-------|----|----|-----|-----|-----|
>  | 0.5B | 1.0 | 0.92 | 0.7 | 0.62 | 0.52 |
>  | 1.5B | 1.0 | 0.91 | 0.87 | 0.8 | 0.79 |
>  | 3B | 1.0 | 0.96 | 0.95 | 0.96 | 0.91 |
>
>  Looking at the number of fingerprints where the relative degradation in utility is 10%, we find that this increases super-linearly with model size (e.g. 0.5B reaches this at ~256 fingerprints, 1.5B at  ~1000 and 3B at 8192) indicating that capacity might increase at a rate better than linear.
>
> ----
> Morris et al. "How much do language models memorize?."
>
> Allen-Zhu et al. "Physics of language models: Part 3.1, knowledge storage and extraction."

---

> > ### Comment · Area_Chair_nJ22 · 2025-08-05
> > **Discussion Required**
> >
> > Dear Reviewer NHAT,
> >
> > Please read carefully through the authors' responses and check if they address all your concerns.
> >
> > With kind regards,
> >
> > Your AC

---

> > ### Comment · Reviewer_NHAT · 2025-08-08
> > **Response to Authors**
> >
> > Thank you for your detailed reply. I found the examples chosen to compare completions between a base model and a finger-printed model quite interesting. If space permits, I would like to see these in the Appendix. Overall, your response has addressed all my questions, and I would like to keep my favorable review.

---

> > > ### Author Response · Authors · 2025-08-08
> > >
> > > Thank you once again for your insightful review. We agree that these qualitative and quantitative results will strengthen the paper, and we will add them to the appendix!

---

### Decision · Program_Chairs · 2025-09-17

**Decision:**

Accept (spotlight)

**Comment:**

Model fingerprinting methods embed hidden "triggers" into a model's behavior, enabling the model owner to subsequently verify their ownership via API queries. This submission presents a novel approach for generating highly scalable, persistent (robust to, e.g., fine-tuning), and harmless (no degradation in model performance) fingerprints. The method’s two main components are: (1) a new fingerprint generation technique called Perinucleus Sampling, and (2) a regularized fine-tuning procedure. The proposed framework can embed a large number of fingerprint key–response pairs while preserving model accuracy. Experimental results show that the method successfully inserts thousands of unique fingerprints with no significant loss in downstream performance, and that these fingerprints remain largely intact even after extensive fine-tuning. All Reviewers agree that the paper is thorough, well-written, and of high quality overall. With respect to content added during the rebuttal, it is recommended that the experiments added during the rebuttal are added to the paper, for example, the comparison between completions from the base model and the fingerprinted model should be included in the Appendix. The additional experiments and clarifications provided during the rebuttal addressed the Reviewers’ concerns. Thus, I recommend the submission for acceptance.